# Characterization of Corrosion Behavior of CLF-1 in Liquid Lithium Using Calibration-Free Laser-Induced Breakdown Spectroscopy in Depth Profile Analysis

**DOI:** 10.3390/ma13010240

**Published:** 2020-01-06

**Authors:** Zhi Cao, Yongtao An, Xianglin Wang, Chang’an Chen, Ying Li

**Affiliations:** 1Key Laboratory of Radiation Physics and Technology, Ministry of Education, Institute of Nuclear Science and Technology, Sichuan University, Chengdu 610065, China; georgecao7@gmail.com; 2Institute of Materials, China Academy of Engineering Physics, Jiangyou 621908, China; terno@126.com (X.W.); chenchangan@caep.cn (C.C.); 3Key Laboratory of Advanced Technology of Materials, Ministry of Education, Superconductivity and New Energy Research and Development Center, Southwest Jiaotong University, Chengdu 610031, China; jierze@163.com

**Keywords:** lithium corrosion, calibration-free laser-induced breakdown spectroscopy, quantitative analysis, depth profile analysis

## Abstract

It is important to get fast and quantitative compositional depth profiles for the boundary layer of the corroded specimen in order to understand the corrosion process and mechanism due to liquid lithium induced corrosion problems to structural material of fusion reactors. In this work, calibration-free laser-induced breakdown spectroscopy (CF-LIBS) is introduced to investigate the compatibility of CLF-1(China low-activation Ferritic steel) exposed in liquid lithium at 500 °C for 500 h. The results show that CF-LIBS constitutes an effective technique to observe the corrosion layer of specimens which are non-uniform and the elements of matrix show gradient distribution from the boundary to the inner layer. The concentration was 82–95 wt.% Fe, 5–12 wt.% Cr, 0.45–0.85 wt.% Mn, 1.6–1.1 wt.% W, 0.11–0.16 wt.% V, and <0.2 wt.% Li along the longitudinal corrosion depth for the corrode CLF-1. The results reveal the quantitative elemental variation trend of CLF-1 in the lithium corrosion process and indicate that the CF-LIBS approach can be applied to the analysis of composition in multi-element materials.

## 1. Introduction

Liquid lithium, as the self-coolant and the tritium breeder applied in Tokamak reactor, has the advantage of high atom density, high heat conductivity. Moreover, its liquid phase could largely avoid the problem caused by thermal expansion and cold shrinkage, as well as radiation damage [1,2]. Thus, the demonstration of stability of structural material against corrosion caused by lithium is crucial [3,4]. There are several major candidate materials for the blanket structure, such as reduced activation ferritic-martensitic (RAFM) [3,4,5], China low-activation martensitic (CLAM) [6], and vanadium-allay. A kind of Chinese RAFM named CLF-1 fabricated by the Southwestern Institute of Physics (SWIP) has good mechanical properties and low sensitivity to radiation-induced swelling, and is considered the structural material candidate for the China Fusion Engineering Test Reactor (CFETR) [7].

The core corrosive problem of structural material against liquid lithium mainly involves the dissolution and penetration of elements [3,8]. In order to investigate the dissolution and penetration of the lithium corrosion process, several technical methods have been considered, such as secondary ion mass spectrometry (SIMS), X-ray photoelectron spectroscopy (XPS), the glow discharge optical emission spectroscopy & mass spectrometry (GD-OES/MS), and inductively coupled plasma mass spectrometry and atomic emission spectrometry (ICP/AES). These techniques require a high vacuum condition, or certain constraining operation pressures, or sample size requirement, otherwise the sample would be destroyed.

Calibration-free laser-induced breakdown spectroscopy (CF-LIBS) has been applied for precious multi-element analysis for various materials, including alloy metal [9], rock analogues, soil [10] and hydrogen isotopes retention at the plasma facing components (PFCs) in the international thermonuclear experimental reactor (ITER) tiles [11]. This approach is qualified to output accurate and continuous chemical compositions of samples without measuring standard curves through certified reference standard samples and internal standards [12,13,14].

In this work, such CF-LIBS technique was developed and used to analyze the trace of penetrated lithium and the distribution of matrix elements concentration in the corroded layer of CLF-1 specimen through depth profile analysis. The corroded layers display different component distribution after being exposed to the liquid lithium at 500 °C for 500 h. The surface of samples became coarse and porous. The results show that the depth profile includes the distinctive longitudinal variations within the corroded layer from interface to the inner substrate, which explains the quantitative elemental variations in the whole static corrosion process. This suggests that CF-LIBS has the capability to demonstrate the relative concentrations of the elements, which could be employed as an efficient approach for the quantitative analysis of lithium corrosion in fusion reactors.

## 2. Materials and Methods

### 2.1. Specimen Preparation

For this experiment, the substrates of CLF-1 with square size of 15 × 15 × 1 mm^3^ were initially prepared. The surfaces of the specimens were mechanically polished and cleaned by alcohol in ultrasonic machine (GT Sonic, Meizhou, China). The chemical composition of specimen is listed in Table 1.

### 2.2. Corrosion Experiment

The substrates were sealed into the Mo crucible in glove box (Vigor, Suzhou, China) with argon atmosphere and then heated at the constant temperature of 500 °C for 500 h. The static corrosion device (Self-developed by Insititue of mateirals, China Academy of Engineering Physics, Jiangyou, China) is made up of a vacuum chamber, pump, heating equipment, and control system. During the corrosion process, the samples were completely immersed in liquid lithium with argon atmosphere. After the experiment, the crucibles were taken out of the device. To avoid the intense reaction with water and to protect the corroded surface, the corroded specimens were cleaned by pure alcohol to remove the adhering lithium until the weight of the specimens remained constant. Through ultrasonic cleaning, all the adhering lithium solved and the whole corroded surface of the specimens were revealed.

### 2.3. Device and Setup of LIBS

The experimental LIBS set-up is shown in Figure 1. It is composed of laser, fiber-optic spectrometer, pulse generators, optical table, and a target holder. LIBS signals were obtained by using a Q-swiched Nd:YAG laser (LTB lasertechnik, Berlin, Germany) with 5 ns pulse width emitting at the fundamental wavelength of 1064 nm. The laser beam was focused onto the substrate surface by a 50-mm-focal-length bi-convex quartz lens.

The analyzing laser beam was directed in a vertical direction to the sample surface which generated a crater with diameter of approximately 0.5 mm. To get high detection sensitivity, a calorimeter was used for adjusting the pulse energy at 70 mJ. After the laser-induced plasma has been set up, the light of the laser-induced plasma was gathered by a quartz collimating mirror (Avants UV-74) which was positioned on an optical table at 60° angle of incidence to the objective. Then, the emission signal was transferred by a long coupling fiber optic (core diameter 400 μm) into an echelle spectrometer with ICCD detector. A delay time was set at 1.0 μs and a gate time was set at 5.0 μs were employed for the plasma emission collection that were satisfied to lower background signals from continuous plasma shots and refrain from intense variations in plasma temperature in the process. In this work, the specimens were ablated by 30 consecutive shots on one spot, and the results were obtained by averaging 5 spectrums. The lines intensities become constant after 15th shoot.

## 3. Results

### 3.1. Microstructure

As shown in Figure 2, the surface of CLF-1 steel before corrosion was smooth and flat, while after exposure in lithium, the surface become rough and porous which would bring the disturbance of Matrix effect for LIBS measurement. Dark spots distributed on the surface have a high content of oxygen and carbon, which maybe formed by the difference of binary phase (Martensitic and prior austenitic) of the CLF-1. The results are shown in Appendix A
Table A1 and Figure A1. The oxygen probably was brought by the impurity from the lithium and the cleaning process.

### 3.2. LIBS Spectra

In order to investigate the elemental dissolution and permeation helping us to understand the liquid lithium corrosion mechanism on structure material CLF-1, CF-LIBS was introduced to analyze the quantitative composition variation along the corroded layer. A sample LIBS spectrum is shown in Figure 3 with a low intensity signal-to-noise ratio. Combined with the National Institute of Standards and Technology (NIST) spectral database and the spectra [15], the influenced elements of the corroded CLF-1, i.e., Fe, Cr, Mn and Li could be clearly distinguished and detected by the characteristic spectrums. The emission lines of W and V with low content in CLF-1 are monitored as well.

#### 3.2.1. Evaluation of Local Thermodynamic Equilibrium and Matrix Effect

In this work, the inhomogeneous samples and the spectral responses of the spectrometer have to be taken into account, which may profoundly affect LIBS measurement. So, it is necessary to estimate the local thermodynamic equilibrium (LTE) conditions of laser-induced plasma and matrix effect to exclude the interference mentioned above [16]. When the LTE and matrix effect conditions are established and valid, the emission line is mostly identified by three parameters: the electron temperature, the electron density, and the elemental concentration [17].

Electron temperatures (Te) are estimated via Boltzmann plots of Fe atomic lines all selected from NIST database as isolated and with weak self-absorption and listed in Table 2. Taking the logarithm of the Boltzmann equation [18], we can represent each Fe I emission line as a point in Boltzmann plot demonstrated in Figure 4. Thus, the line intensity is figured as a Boltzmann plot where the slope of the lines is related to the electron temperature which fluctuates around 8000 K, as shown in Figure 5.

Based on McWhirter’s criterion [19], the necessary condition of LTE is determined by the electron density (*Ne*). Thus, to calculate electron density, Fe I line at 426.047 nm is chosen to measure *Ne* because of the good SNRs and no other overlapping spectral lines. The Stark broadening of Fe I 426.047 nm expressed by full width at half maximum (FWHM) is fitted by Lorentzian functions [20,21] as follows:(1)$$\Delta \lambda_{1/2}=2\omega \cdot \left(N_{e}/10^{16}\right)$$
where Δ*λ*_1/2_ is the FWHM of the emission line, *Ne* is the electron number density of plasma, and ω is the electron width parameter. The *Ne* values in the range of 0.5–1.5 × 10^17^ cm^−3^ as function of LIBS shots is shown in Figure 5.

Based on the McWhirter criterion [17], the critical lower limit of *Ne* (cm^−3^) for LTE can be described by:(2)$$Ne\geq 1.6\times 10^{12}T^{\frac{1}{2}}{\left(\Delta E\right)}^{3}$$
where T (K) refers to the plasma temperature, and ΔE (eV) the energy transition gap in this study equals to 2.18 eV [16]. The measured value of electron densities 1.5 × 10^15^ cm^−3^ is two orders lower than the electron densities measured by the stark broadening. Therefore, the value of *Ne* meets with the McWhirter conditions that establish the LTE conditions.

Figure 5 presents the curves with error bar of electron temperature and electron density for laser induced plasma. The relative standard deviation (RSD) of plasma temperature and electron density are 7.8% and 18.5%, respectively. For LIBS measurement, it is justified which the matrix effect couldn’t be ignored [22].

#### 3.2.2. Chemical Depth Profile of CF-LIBS Results

When the LTE condition is met, the CF-LIBS method advises that the line integral intensity corresponding to elemental concentration can be expressed as [14]:(3)$$\bar{I_{\lambda }^{ki}}=FC_{s}A_{ki}\frac{g_{k}e^{-\left(E_{k}/K_{B}T\right)}}{Q_{s}\left(T\right)}$$
where $\bar{I_{\lambda }^{ki}}$ is the measured integral line intensity, $Q_{s}\left(T\right)$ is the partition function, *C_s_* is the concentration of the emitting atomic element, $A_{ki}$ is the transition probability, *g_k_* and *E_k_* are the energy and degeneracy of the level retrieved from the NIST database and *F* is the parameter which determines the optical efficiency of the acquisition system, like the plasma energy and density. In the whole experimental process, it is important for the stability of the *F* constant to remain the within the experimental parameters fixed in a certain measurement range, such as laser energy, focus, optical path, and so on.

The logarithm of Equation (3) yields a mathematical method for the determination of electron temperature. Thus, the line intensity can be graphically represented as a Boltzmann plot where the slope and intercept ($q_{s}$) of the straight line is correlated with the electron temperature and the elements concentration, respectively. In the LTE theory, once the electron temperature is determined, the concentration of the element can be measured by the calculation of the spectral intensity. *F* could be identified by the unitary sum of the element concentration *C_s_* [14].
(4)$$\sum_{S}C_{s}=\frac{1}{F}\sum_{s}Q_{s}\left(T\right)e^{q_{s}}=1$$

The concentration of all the elements in specimens can be calculated from:(5)$$\;\;\;\;\;C_{s}=\frac{Q_{s}\left(T\right)}{F}e^{q_{s}}$$

To optimize the results and accuracy, the selection of the spectral line has to consider other several parameters. First, the neutral and singly ionized lines with excited lower state are preferred over the ground state to avoid the self-absorption. Second, we calculate the mean value of spectrum by averaging the line intensities of each laser pulse respectively for five times. The selected spectral lines, listed in Table 3, have good SNRs and weak self-absorption, indicating that those typical matrix elements as well as Li have the good qualification to be detected. In this experiment, a number of 15 spectra are consecutively recorded to demonstrate the quantitative elemental content with their intensities. The concentration curve variations as a function with pulse number may reflect the mechanism of the elemental transaction in the depth profile during the corrosion process, shown in Figure 6.

In the liquid metal corrosion mechanism, the corrosion medium lithium permeates into the crystal boundary phase by thermal diffusion and mass transfer with the substrate compositions [23]. Figure 6 shows that the quantitative content curve of Li penetrated into the substrate from the intersurface to the inner matrix. The concentration of Li contents decreases dramatically from 0.2 wt.% to the almost zero. At 12th shot, the concentration of Li is close to zero. The area from the first shot to the 12th shot represents the permeated depth of lithium, and we consider this area as a corrosion layer.

Since Fe and Cr are the main components, their relative concentrations would affect each other. The concentration of Fe decreases from 94% to 84% in the corrosion layer. The concentration fluctuations is due to the binary phase (martensite and austenite) of CLF-1, where lots of Fe-C compounds form segregation in high temperature corrosion [23,24,25,26]. In the case of Cr, it is clearly demonstrated that the concentration of Cr fluctuates from 5% to 12% at the first six shots, then tend to be stable at 8.5%. During the corrosion process, Cr present in the crystal boundary in a form of carbon compounds (M_23_C_6_, M = Cr, V, W) which is exchanged by Li in the diffusive and dissovled process [27,28]. In the meantime, the spread speed of Cr in the crystal grain is relatively slow causing the reduction of Cr in the grain boundary that cannot be replenished in time. In comparison, the relative concentration of Cr dissolution is higher than Fe which may be attributed to higher diffusion coefficient of Cr to Fe in liquid Li [29]. In the case of Mn, the diffusion rate of Mn from the inner layer to the interface is quick in exposure to liquid lithium. When Mn in the interface dissolved into the liquid lithium, Mn transfer from the interior to the interface made the concentration of Mn of up to 0.8%. The concentration of V increases from 0.11% to 0.16% along the corrosion layer. V is sensitive to the exchange of non-metals like nitrogen that induce the reduction of concentration near the surface regions [30,31]. The calculated concentration of V before and after the corrosion was lower than the concentration of 0.26%, which could be caused by the relatively low content in the substrate. Further, for the element W, because of the solution of W in liquid lithium is low [31], the concentration change of W is not evident, and decreases from 1.6% to 1.2%.

## 4. Conclusions

The structural material CLF-1 has been exposed to liquid lithium at 500 °C for 500 h. The surface of CLF-1 became coarse and trace lithium was found in the surface. The calibration-free LIBS method was introduced to perform quantitative elemental composition depth profiles analysis for CLF-1 exposed to liquid lithium. The results demonstrate the ability of CF-LIBS for in-depth chemical composition analysis of the corrosive specimens and obtained different element distributions in the corrosion layer, including trace element and corrosion medium Li. It should be noted that this method would produce scattered values because of highly inhomogeneous corroded samples and binary-phase which made the sample composition segregated.

From the surface to the inner layer, the concentration of Fe, Mn, W, and Li decreased from 95% to 83%, 0.85% to 0.45%, 1.6% to 1.1%, and 0.2% to almost 0, respectively. Cr increased from 0.11% to 0.16%. V fluctuated between 5% and 12%. The elemental variation in the depth profile represents a considerable index to explain the corrosion resistance of steels within liquid lithium. This study shows that the CF-LIBS measurement represents a useful procedure for the quantitative measurement of the elemental variation of corroded structural materials in the fusion blankets. It provides a fast, remote, real-time, and convenient way to monitor the compatibility of the liquid metal blankets for fusion reactor applications.

## Figures and Tables

**Figure 1 materials-13-00240-f001:**
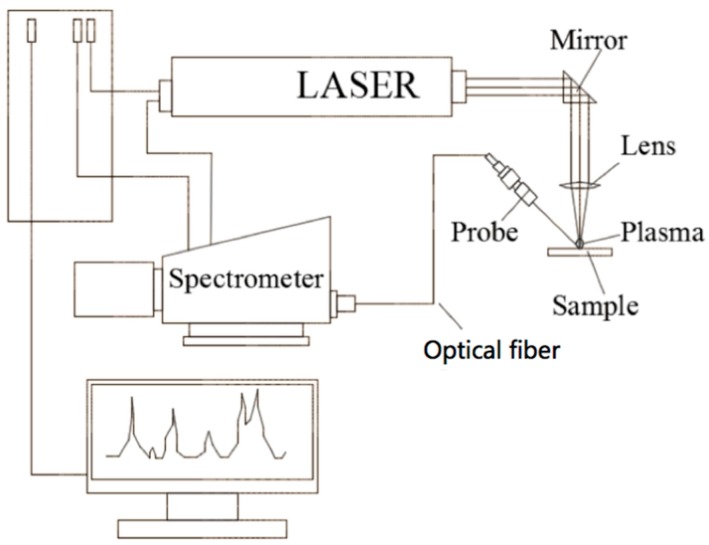
Diagram of Laser-Induced Breakdown Spectroscopy (LIBS) setup.

**Figure 2 materials-13-00240-f002:**
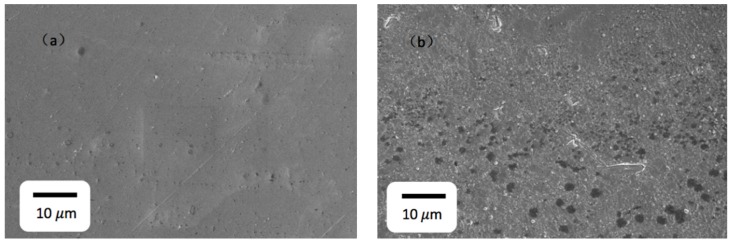
SEM images of CLF-1 (**a**) before liquid lithium corrosion; (**b**) after liquid lithium corrosion.

**Figure 3 materials-13-00240-f003:**
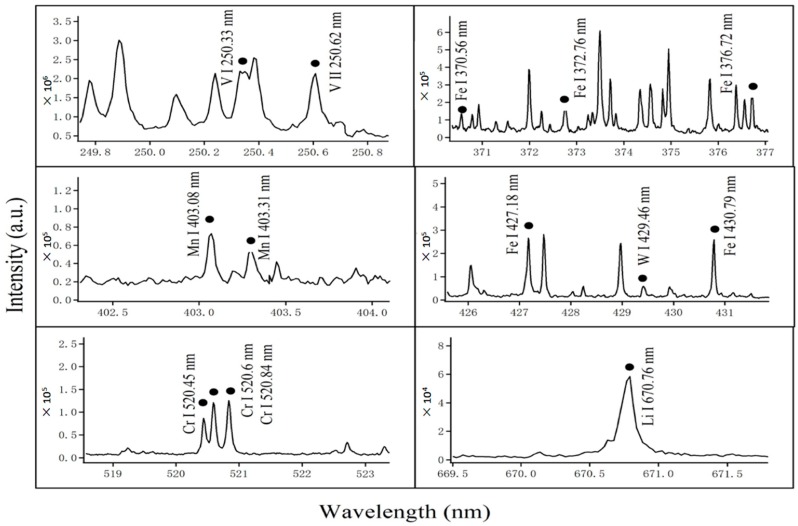
Typical LIBS spectra of CLF-1 steels after exposure to Li.

**Figure 4 materials-13-00240-f004:**
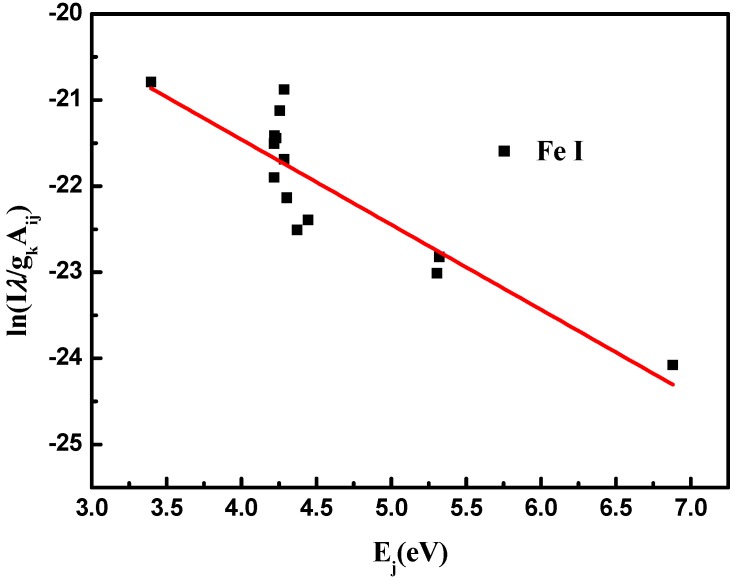
Boltzmann plot for Fe I lines.

**Figure 5 materials-13-00240-f005:**
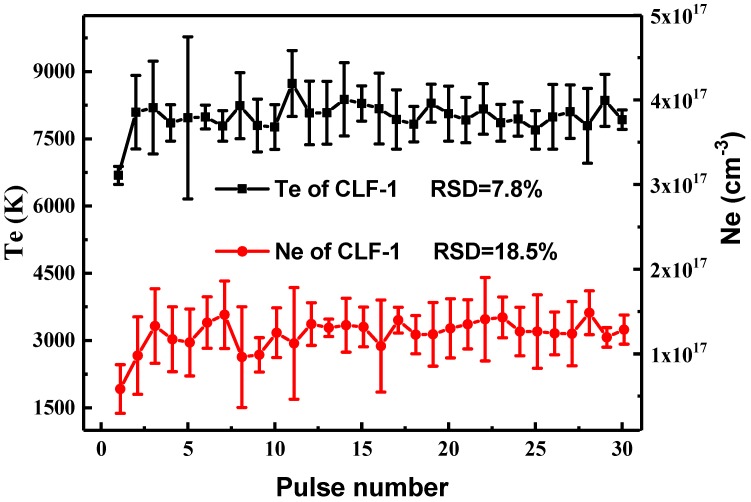
Depth profiles of the excitation temperature and the electron density on the CLF-1 steel surface.

**Figure 6 materials-13-00240-f006:**
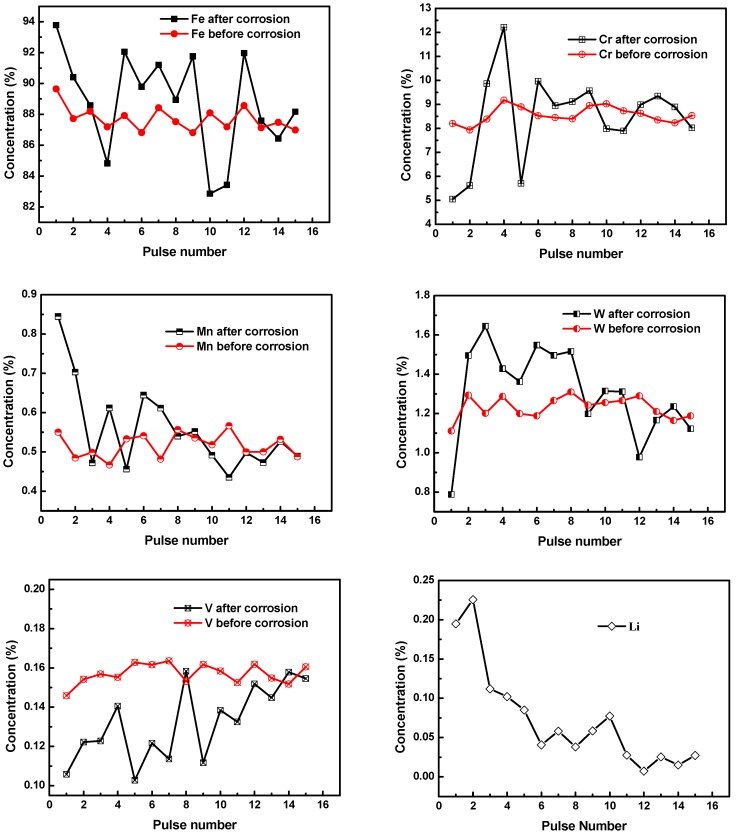
Depth profiles of elemental quantitative concentration normalized by CF-LIBS method for CLF-1 before and after exposure to liquid lithium. The pulse numbers approximately represent the value of depth.

**Table 1 materials-13-00240-t001:** Chemical compositions of the China low-activation Ferritic steel (CLF-1) (wt.%).

Sample	Cr	Mn	C	W	V	Ta	N	Fe
CLF-1	8.5	0.5	0.11	1.5	0.26	0.1	0.03	Balance

**Table 2 materials-13-00240-t002:** Fe I lines and parameters for Boltzmann plot method.

Species	Wavelength (nm)	*A_ij_* (s^−1^)	*g_k_*	*E_i_* (eV)	*E_j_* (eV)
Fe I	277.390	9.36 × 10^7^	9	2.4534	6.8744
	281.329	3.42 × 10^7^	11	0.9146	5.3204
	360.886	8.13 × 10^7^	5	1.0111	4.4456
	363.146	5.17 × 10^7^	9	0.9582	4.3714
	370.557	3.21 × 10^6^	7	0.0516	3.3965
	372.762	2.24 × 10^7^	5	0.9582	4.2833
	376.719	6.39 × 10^7^	3	1.0111	4.3013
	378.788	1.29 × 10^7^	5	1.0111	4.2833
	379.500	1.15 × 10^7^	7	0.9901	4.2562
	425.079	1.02 × 10^7^	7	1.5574	4.4733
	426.047	3.99 × 10^7^	11	2.3992	5.3085
	427.176	2.28 × 10^7^	11	1.4849	4.3865
	430.790	3.38 × 10^7^	9	1.5574	4.4346
	438.354	5.00 × 10^7^	11	1.4849	4.3125
	440.475	2.75 × 10^7^	9	1.5574	4.3714

**Table 3 materials-13-00240-t003:** Selected emission lines for depth profile analysis.

Species	Wavelength (nm)	*A_ij_* (s^−1^)	*g_k_*	*E_i_* (eV)	*E_j_* (eV)
V I	250.330	4.40 × 10^7^	4	0.0000	4.9513
	306.638	2.10 × 10^8^	12	0.0685	3.9597
	318.398	2.50 × 10^8^	5	0.0000	3.1758
	438.472	1.10 × 10^8^	8	0.0000	3.0751
V II	250.622	9.72 × 10^7^	9	1.0962	6.0418
	309.310	2.00 × 10^8^	13	0.3921	4.3994
W I	386.798	4.60 × 10^6^	3	0.0000	3.8807
	429.460	1.24 × 10^7^	3	0.0000	3.2633
Mn I	279.480	3.70 × 10^8^	58	0.0000	4.4349
	403.076	1.65 × 10^7^	6	0.0000	3.0751
	403.307	1.58 × 10^7^	4	0.0000	3.0733
	403.450	1.58 × 10^7^	4	0.0000	3.0722
	404.141	7.87 × 10^7^	10	2.1142	5.1812
Mn II	257.615	2.80 × 10^8^	9	0.0000	4.8114
	294.928	1.96 × 10^8^	7	1.1745	5.3772
	348.870	2.11 × 10^7^	3	1.8475	5.4004
Cr I	425.440	3.15 × 10^7^	9	0.0000	2.9134
	427.480	3.07 × 10^7^	7	0.0000	2.8995
	428.970	3.16 × 10^7^	5	0.0000	2.8894
	520.450	5.09 × 10^7^	3	0.9414	3.3230
	520.602	5.14 × 10^7^	5	0.9414	3.3223
	520.841	5.06 × 10^7^	7	0.9414	3.3212
Cr II	284.980	9.20 × 10^7^	8	1.5100	5.8622
	286.090	6.90 × 10^7^	4	1.4800	5.8245
	286.263	6.30 × 10^7^	8	1.5300	5.8683
	286.671	1.20 × 10^8^	4	1.4918	5.8154
	311.865	1.70 × 10^8^	4	2.4211	6.3956
	312.041	1.50 × 10^8^	6	2.4339	6.4061
Li I	610.354	5.71 × 10^7^	4	1.8478	3.8786
	670.776	1.47 × 10^8^	4	0.0000	1.8478
Fe I	360.886	8.13 × 10^7^	5	1.0111	4.4456
	363.146	5.17 × 10^7^	9	0.9582	4.3714
	370.557	3.21 × 10^6^	7	0.0516	3.3965
	372.762	2.24 × 10^7^	5	0.9582	4.2833
	376.719	6.39 × 10^7^	3	1.0111	4.3013
	378.788	1.29 × 10^7^	5	1.0111	4.2833
	379.500	1.15 × 10^7^	7	0.9901	4.2562
	400.524	2.04 × 10^7^	5	1.5574	4.6520
	404.581	8.62 × 10^7^	9	1.4849	4.5485
	407.174	7.64 × 10^7^	5	1.6079	4.6520
	425.079	1.02 × 10^7^	7	1.5574	4.4733
	426.047	3.99 × 10^7^	11	2.3992	5.3085
	427.176	2.28 × 10^7^	11	1.4849	4.3865
	430.790	3.38 × 10^7^	9	1.5574	4.4346
	438.354	5.00 × 10^7^	11	1.4849	4.3125
	440.475	2.75 × 10^7^	9	1.5574	4.3714
Fe II	261.187	1.20 × 10^8^	8	0.0477	4.7932
	261.382	2.12 × 10^8^	2	0.1069	4.8489
	262.166	5.60 × 10^7^	2	0.1211	4.8489
	262.566	3.52 × 10^7^	10	0.0477	4.7683
	262.829	8.74 × 10^7^	4	0.1211	4.8370
	263.104	8.16 × 10^7^	6	0.1069	4.8178
	266.466	1.91 × 10^8^	10	3.3866	8.0381
	268.475	1.57 × 10^8^	10	3.8143	8.4310
	298.554	2.39 × 10^7^	4	1.7239	5.8755

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
