# Peer review of "Characterization of Corrosion Behavior of CLF-1 in Liquid Lithium Using Calibration-Free Laser-Induced Breakdown Spectroscopy in Depth Profile Analysis"

_materials, 2020, doi:10.3390/ma13010240_

Round 1

Reviewer 1 Report

Authors are presenting a systematic work on the corrosion characteristics into low-activation Ferritic steel composite by using calibration free LIBS method. This work would be useful for the scientific and industrial community.

I would recommend the publication after a minor revision.

I would like to thank all the Authors for their efforts and I kindly ask them to address my comments and suggestions below:

English language and style are fine still some paragraphs are not clearly written and well presented. Some editing is required. The calibration Free LIBS is a well-established quantification method. To this content the use of CF-LIBS is not a novel technique for depth profile analysis. I kindly ask the authors enhance the introduction part and better express the novelty of their work. The used abbreviations must be clearly identified such as CLAM, RAFM, CFTER, V-allay, PFCs , ITER vs… Also, some notations must be identified clearly. For instance “Bal.” in table 1. Line 60-71:
“After the experiment, the corrosion specimens were taken out and cleaned by an alcohol solution until the weight of the specimen remained constant.”
This phrase is not very clear; the sample preparation method should be better explained together with the physical changes occurring during the process if there is any. Line 80-81
“The analyzing laser beam was directed in a vertical direction to the sample surface which generated a diameter of approximately 0.5 mm crater.” What is meant is not so clear. The crater depth and/or the spot size? Line 88-90:
“In this work, the specimens were ablated by 30 consecutive shots on each same site in a direction perpendicular to the specimen surface. The lines intensities become constant after 15th shoot”.
I would suggest the authors to enlarge the plot in Figure-5. up to 30th shoots to better show this concept. In Figure-4. The FE I lines with Ej around 4-5 (eV) there is a large deviation in the Boltzmann plot. I kindly ask the authors to comment on the possible reasons of deviation in terms of the LTE condition. Moreover, the given points in the Boltzmann plot are not totally listed in the Table-2. A specially the given point at around 7 eV, one of the points around 5 eV and some of the point around 4 eV. Line 95-97:
“Dark spots distributed on the surface are high content of oxygen and carbon which maybe formed by the difference of binary phase (Martensitic and prior austenitic) of the CLF-1.”

I kindly ask the authors to better identify what is meant by the “high content of oxygen”. To better clarify the possible oxidation states and/or the formation of carboxylic(if so) groups observed by including references. And I also suggest the authors to give a brief explanation of the Martensitic and prior austenitic phases and how these phases are correlated with their samples. Line 240-241:
“It should be noted that this method would produce values fluctuation because of highly in-homogeneous corroded samples and binary-phase which made the sample composition segregation.” To this content I kindly ask the authors to include the standard deviation and/or the limit of quantification within the data plots.

Reviewer 2 Report

line 14: replace 'because' with 'due to'
line 31: replace 'and either for' with 'as well as'
line 32: replace 'While' with 'Thus'
line 37: replace 'are' with 'is'
line 50: replace 'developed has been' with 'has been developed and'
line 53: insert 'being' between 'after' and 'exposed'
line 63: replace 'compositions of specimens' with 'composition of specimen is'
line 73: replace 'showed' with 'shown'; remove 'Basically,' and start sentence with 'It'
line 81: move 'crater' in front of 'diameter' and add 'with' so it reads 'a crater with'
line 83: insert 'and' before 'collected'
line 94: replace 'disturb' with 'disturbance'
line 104: replace 'specific' with 'sample'
line 110: insert 'after' between 'steels' and 'exposure'
line 113: replace 'magnificently' with 'profoundly'
line 114: ')' is positioned off-line
line 118-120: I would rewrite this whole sentence as:
'Electron temperatures (T_e) are estimated via Boltzmann plots of Fe atomic lines all selected from NIST database as isolated and with weak self-absorption and listed in Table 2."
line 123: replace 'fluctuate at 8000K' with 'fluctuates around 8000K as'
line 129: replace 'choose' with 'chosen'
line 137: replace 'Base' with 'Based'; 'McWhriter' should be 'McWhirter'
line 146: replace 'is' with 'are'
line 147: add ', respectivelly.' after '21.9%'; replace 'it's' with 'it is'
line 168: replace 'meet' with 'is met'; replace 'defines' with 'advises'
line 171: reolace 'the same' with 'yields'; replace 'as' with 'for determination of'
line 174: add 'is' after 'temperature'
line 176: replace 'normalized to unit the' with 'unitary'
line 180: Eq.(5) seems incomplete; 'qs' is nowhere defined, not even in Eq(4) where it appears for the first time;
line 181: there is no '\bar{I_{lambda}^{kl}}' in Eq.(5);
line 182: there are no 'A_{ki}', 'g_{k}', nor 'E_{k}' in Eq.(5)
line 183: there is no 'F' in Eq(5), and 'an' should be 'the'
line 184: 'determined' should be 'determines'
line 185-187: I would add here at least an estimate with uncertainty for the 'F' value as it pertains to your experimental set up
line 189: replace 'neutral emission' with 'neutral and singly ionized'; replace 'with lower' by 'with excited lower'; replace 'to choose rather' with 'over'
line 190: delete 'than'; replace ', only including neutral lines I and single ionized lines II' with fullstop '.'
line 204: replace 'exposed' with 'exposure'
line 211: replace 'is' with 'as'
line 212: replace 'Fe and Cr as the main component, the' with 'Since Fe and Cr are the main components, their'
line 213: replace 'corrode' with 'corrosion'; replace 'The concentration fluctuated' with 'The likely couse for concentration fluctuations'
line 214: replace 'dramatically that may be caused by' with 'is due to'
line 217: replace 'presented into' with 'present in'
line 220: replace 'that cause' with 'causing'; replace 'can not' with 'that cannot'
line 225: replace 'dissolute' with 'dissolve'
line 226: replace 'inner' with 'interior'; delete 'which'; replace 'Mn is' with 'Mn of'
line 232: replace ',decrease' with ' and decreases'
line 234: replace 'is carried out exposure' with 'has been exposed'
line 240: replace 'values fluctuation' with 'scattered values'
line 241: replace 'segregation' with 'segregated'
line 245: replace 'The' with 'This study'
line 246: delete 'rsearch'
line 248: delete 'using life related to the'
line 249: replace 'with' by 'of'; replace 'relate to' with 'blankets for'; replace 'reactor.' with 'reactor applications.'
